# Evidence-Based View of Safety and Effectiveness of Prokineticin Receptors Antagonists during Pregnancy

**DOI:** 10.3390/biomedicines9030309

**Published:** 2021-03-17

**Authors:** Deborah Reynaud, Frederic Sergent, Roland Abi Nahed, Wael Traboulsi, Constance Collet, Christel Marquette, Pascale Hoffmann, Gianfranco Balboni, Qun-Yong Zhou, Padma Murthi, Mohamed Benharouga, Nadia Alfaidy

**Affiliations:** 1Institut National de la Santé et de la Recherche Médicale U1292, Biologie et Biotechnologie pour la Santé, 38000 Grenoble, France; deborah.reynaud.89@gmail.com (D.R.); Frederic.sergent@cea.fr (F.S.); rolandabinahed@gmail.com (R.A.N.); constance.collet@cea.fr (C.C.); christel.marquette@cea.fr (C.M.); PHoffmann@chu-grenoble.fr (P.H.); 2Commissariat à l’Energie Atomique et aux Energies Alternatives (CEA), Biosciences and Biotechnology Institute of Grenoble, 38000 Grenoble, France; 3Service Obstétrique & Gynécologie, Centre Hospitalo-Universitaire Grenoble Alpes, University Grenoble-Alpes, CEDEX 9, 38043 Grenoble, France; 4Lombardi Comprehensive Cancer Center, Laboratory for Immuno-Oncology, Georgetown University Medical Center, Washington, DC 20057, USA; wt247@georgetown.edu; 5Department of Life and Environmental Sciences, University of Cagliari, 09124 Cagliari, Italy; bbg@unife.it; 6Department of Pharmacology, University of California, Irvine, CA 92697, USA; qzhou@uci.edu; 7Monash Biomedicine Discovery Institute, Monash University, Clayton, VIC 3168, Australia; padma.murthi@monash.edu; 8Department of Obstetrics and Gynecology, the University of Melbourne, Parkville, VIC 3010, Australia

**Keywords:** prokineticin antagonists, EG-VEGF, pregnancy pathologies, therapy, angiogenesis, trophoblast invasion

## Abstract

Endocrine gland derived vascular endothelial growth factor (EG-VEGF) is a canonical member of the prokineticin (PROKs) family. It acts via the two G-protein coupled receptors, namely PROKR1 and PROKR2. We have recently demonstrated that EG-VEGF is highly expressed in the human placenta; contributes to placental vascularization and growth and that its aberrant expression is associated with pregnancy pathologies including preeclampsia and fetal growth restriction. These findings strongly suggested that antagonization of its receptors may constitute a potential therapy for the pregnancy pathologies. Two specific antagonists of PROKR1 (PC7) and for PROKR2 (PKRA) were reported to reverse PROKs adverse effects in other systems. In the view of using these antagonists to treat pregnancy pathologies, a proof of concept study was designed to determine the biological significances of PC7 and PKRA in normal pregnancy outcome. PC7 and PKRA were tested independently or in combination in trophoblast cells and during early gestation in the gravid mouse. Both independent and combined treatments uncovered endogenous functions of EG-VEGF. The independent use of antagonists distinctively identified PROKR1 and PROKR2-mediated EG-VEGF signaling on trophoblast differentiation and invasion; thereby enhancing feto-placental growth and pregnancy outcome. Thus, our study provides evidence for the potential safe use of PC7 or PKRA to improve pregnancy outcome.

## 1. Introduction

Prokineticin family includes two conserved small proteins, prokineticin 1 (PROK1) and prokineticin 2 (PROK2) [1,2]. Due to its similarities in action to VEGF (Vascular Endothelial Growth Factor) [1], PROK1 is also called EG-VEGF (Endocrine Gland derived-VEGF). While VEGF has mainly been reported to play a crucial role in promoting angiogenesis with specific and strong mitogenic and chemotactic actions on endothelial cells and induces microvascular permeability [3], prokineticins act as both pro-angiogenic and pro-inflammatory factors [1,4,5,6,7]. They bind to two highly related G protein-coupled receptors (GPCRs), prokineticin receptor 1 (PROKR1) and prokineticin receptor 2 (PROKR2) [6,7]. Since their discovery, prokineticins and their receptors have been reported to be expressed in a wide range of tissues [2,8,9,10]. While, PROK1 is predominantly expressed in the peripheral tissues, especially steroidogenic organs [4,9,11,12,13], PROK2 is mainly expressed in the central nervous system and non-steroidogenic cells of the testes [14,15,16,17]. Prokineticin family has been implicated in several important physiological functions, including gastrointestinal smooth muscle contraction, circadian rhythm regulation, neurogenesis, angiogenesis, pain perception, mood regulation, and reproduction [8,11,12,15,16,17,18,19,20,21,22]. Dysregulation of the prokineticin signaling has been observed in a variety of diseases, such as cancer, ischemia, neurodegeneration, abnormal angiogenesis and pregnancy pathologies [14,23] 

In relation to pregnancy, we and others have recently demonstrated that EG-VEGF controls key processes of human placental development [4,12,18,20,24,25,26,27,28]. EG-VEGF exhibits a dynamic expression in the placenta, with a peak of expression during the first trimester of pregnancy, 8–11 weeks of gestation [4,20,29]. At the cellular level, EG-VEGF is mainly expressed in the syncytiotrophoblast (ST), and in hofbauer cells [4,20,29]. During the first trimester of pregnancy, EG-VEGF increases trophoblast proliferation via PROKR1 and controls their precocious invasion via its PROKR2 receptor [4,20,29]. EG-VEGF also enhances angiogenesis within the placental villi through both receptors [4]. 

In relation to pregnancy pathologies, we have demonstrated that EG-VEGF and its receptors are increased in the most threatening pathology of pregnancy, preeclampsia (PE) as well as, fetal growth restriction (FGR), and recently in gestational choriocarcinoma [26,27,28,29,30,31,32], suggesting that the antagonization of EG-VEGF signaling may constitute a therapeutic promise in pregnancy pathologies.

During the last decade, two specific antagonists for PROKR1 and PROKR2 have been developed [15,33]. These antagonists are reversible non-peptide molecules that have also been shown to reverse prokineticin-mediated processes, such as inflammation, arthritis, chronic pain and memory impairment [13,34,35,36,37].

In relation to pregnancy, we have recently demonstrated that PROKR1 and PROKR2 antagonists reversed EG-VEGF mediated proliferation, migration and invasion of tumor trophoblast cells [6]. These findings strongly support the potential use of these antagonists to reverse prokineticin signaling in pregnancy pathologies such as PE and FGR. Nevertheless, their potential adverse effects on the pregnancy outcome, in vivo, have never been investigated.

In previous studies using gravid mice, we demonstrated that both EG-VEGF and its receptors exhibit similar profiles and patterns of expression in mouse placenta, as in human. EG-VEGF expression peaks during the first 11.5 days post coitus (dpc), equivalent to the first trimester in pregnant women and is localized in the ST within the labyrinth. Its receptors are expressed in the same cell types and are mainly expressed during the first-second trimesters [12,25,38]. These finding strongly support the use of the gravid mouse as a model to determine the impact of both antagonists on placental development and on the pregnancy outcome.

Here, we conducted an in vivo and in vitro studies to characterize the independent and combined effects of PROKR1 and PROKR2 antagonists on pregnancy outcome in gravid mice and in the rodent invasive trophoblast cells, the RCHO-1 cells.

## 2. Materials & Methods

### 2.1. Animals 

All animal studies were approved by the institutional guidelines and those formulated by the European Community for the Use of Experimental Animals. Our study was approved under the following ethical number: APAFIS# 14176-2018032016534632v1 (28 August 2018). Swiss mice were housed under controlled illumination (12:12 h light: dark cycle). Food was available ad libitum. Two to three months-old mice were mated and used to generate gravid mice. The date of the presence of a vaginal plug was taken as 0.5 day post coitus (dpc). The gravid mice were randomly assigned to receive at days 4.5 dpc, 7.5 dpc and 10.5 dpc one or a combination of PROKR1 (PC7) and PROKR2 (PKRA) antagonists. PC7, is a non-peptide PKOKR1-preferring antagonist that was obtained from Dr Balboni [33,39]. PC7 was reported to be 10 times more potent than the reference PC1 [40] and was used at [0.5 mg/kg] and injected, subcutaneously. PKRA is a non-peptide PROKR2-preferring antagonist that was obtained from Dr Zhou [15]. PKRA was used at [75 mg/kg] and delivered subcutaneously as previously reported [15,28,41]. In total, 31 mice were used at 12.5 dpc: Controls for PC7 and PKRA (*n* = 7 mice); PC7 (*n* = 6 mice), PKRA (*n* = 6 mice), controls for PC7+PKRA (*n* = 6 mice) and PC7 + PKRA (*n* = 6 mice). Because combined PC7 and PKRA were injected separately, the control mice for the combined treatments also received two injections. Hence, the controls for the independent or combined treatments were considered as different. Mice were sacrificed at 12.5 dpc (see protocols of collection on Figure 1). The blood was drawn by cardiac puncture before laparotomy. All placentas and fetuses were weighed and stored at −80 °C or fixed for further analyses. Average weights were analyzed as raw data. Placental efficiencies were determined by calculating the fetal/placental weights ratios.

### 2.2. Real-Time RT-PCR Analysis of Placental Tissue

Total RNAs were extracted from placentas using phenol supplemented-Trizol (Sigma-Aldrich, Saint Quentin Fallavier, France) and precipitated by chloroform after 20 min (12,000× *g* at 4 °C) centrifugation. First-strand cDNAs were synthesized from 1 µg of total RNA by reverse transcription using the iScript system (BioRad, Marnes-la-Coquette, France), according to the manufacturer’s instructions. Quantitative polymerase chain reaction (RT-qPCR), using SYBER-green, qPCR Master Mixwas (Promega, Charbonnières-les-Bains, France) was performed on a Bio-Rad CFX96 apparatus. Relative quantification of Proliferin, Placental lactogen 1, Placental lactogen 2, Hand 1, Mash 2, Gcm1 gene expression was normalized to GAPDH mRNA levels. Sequences of the PCR primers used are listed in Table 1.

### 2.3. Placental Histology

Following overnight fixation in 10% (*v*/*v*) neutral buffered formalin (Sigma-Aldrich, France) or accustain FFF (formalin free tissue fixative) at 4 °C, tissues were preserved in ethanol 70% until embedding in paraffin and sectioning (5 μm).

### 2.4. Periodic Acid-Schiff Staining

Sections were stained with periodic acid-Schiff (PAS) (Sigma Aldrich, France) to identify glycogen cells. Slides were counterstained using hematoxylin (Sigma Aldrich, France). At least three sections per placenta, collected from each animal, were analyzed.

### 2.5. Immunofluorescence

Placental tissues were incubated with mouse podocalyxin (10 µg/mL) (R &D system, France) primary antibody. Incubation was performed overnight at 4 °C in PBS with 2% goat serum, 1% bovine serum albumin. Slides were then washed and incubated with a secondary antibody (Goat-antimouse Cy3, Abcam) for 1 hour at room temperature and counterstained with Hoechst 33342. Images were taken by Zeiss AxioVision microscope, and processed using AxioVision SE64 Rel. 4.9.1 software [28]. Negative controls were treated in an identical manner, except that podocalyxin antibody was replaced by PBS.

### 2.6. Western Blotting Analyses of Placental Tissues 

Total placental protein was extracted by mechanically homogenizing placentas on ice for 3 min in RIPA lysis buffer [50 mm Tris-HCl (pH 7.5), 150 mM NaCl, 1% sodium deoxycholate, 0.1% sodium dodecyl sulfate, 1% Triton X-100, 1 mM phenylmethylsulfonylfluoride, 5 µg/mL leupeptin, 5 μg/mL aprotinin + 1% proteases inhibitors]. The homogenates were centrifuged (14,000× *g* at 4 °C) for 20 min, and the supernatants were collected. Protein concentration was determined using the Bradford assay. Samples were diluted in miliQ water and read at 595 nm wavelength. 20 to 40 μg of protein extracts were electrophoretically separated on Biorad Precast gels (BIORAD, Mini-PROTEAN TGX, stain free 4–15%) for immunoblot analysis using the following antibodies mouse anti-PCNA (2 µg/mL) (Abcam, Paris, France), anti-Carbonic Anhydrase IX (CA9, 5 µg/mL) (Novus Biological, Lille, France), anti-CD31 (BD, France). Protein transfer was performed using the rapid Biorad device (TRANS-BLOT TURBO, program: MIXED MW 7 min–25 V). The blots were incubated with biotinylated goat anti-rabbit IgG (450 ng/mL, 1:2000) or biotinylated goat anti-mouse IgG (450 ng/mL, 1:5000 in blocking solution) for 1h. Antibody-antigen complexes were detected using the ECL plus detection system (BioRad, Marnes-la-Coquette, France). β-actin was used as loading control to normalize the total protein load in each sample.

### 2.7. RCHO-1 Cell Line Culture

For in vitro studies, we used the rat trophoblast cell line RCHO-1. The RCHO-1 cell line provides an effective in vitro model system for dissecting the trophoblast cell differentiation pathway, as they exhibit many characteristics of trophoblast stem cells [42,43]. 

There are two strong advantages for the use of these cells. First, RCHO-1 is a rodent cell line, and second, this cell line can be maintained in a proliferative (i.e., stem cells) or differentiated state (i.e., giant cells). RCHO-1 cells maintain their proliferative potential when cultured in RPMI 1640 medium supplemented with 20% fetal bovine serum (heat inactivated), 100 mg/mL penicillin-streptomycin, 1 mM sodium pyruvate, and 50 mM 2-mercaptoethanol in a 37 °C incubator under 95% air-5% CO_2_. Three days after the cells were cultured under proliferative conditions, a differentiated state could be obtained by switching to RPMI 1640 medium containing 10% horse serum [42,43]. Trophoblast giant cell differentiation was verified by the morphological detection of trophoblast giant cells [42,43]. RCHO cells were treated with the antagonists alone or in combination. Same control was used for all conditions as final working concentrations for both antagonists were diluted in PBS. Both antagonists were used at a concentration of 1 µM. This concentration was chosen according to previous reports on the use of PROKR1 or PROKR2 antagonists in vitro [28,44,45].

### 2.8. Wound-Healing Assay

RCHO-1 cells were seeded and grown to confluency in 12 well plates. Differentiated confluent RCHO-1 cultured cells (i.e., giant cells) in a dish were scratched with a sterile tip to create an artificial wound which was allowed to heal for the next 12 h. The cells were treated at T0 with the antagonists alone or in combination at 1 µM. in the presence of mytomicin (1 µM) to inhibit proliferation during the wound closure. The treatment lasted for 12 h. The size of the wound was measured and quantified using the photomicrographic images captured from three separate experiments. The closing of the wound was analyzed using Scion Image software (version 4.0.2). The results are presented as percentage of wound closure following 12 h of treatment. 

### 2.9. Matrigel Invasion Assay

Cultured RCHO-1 cells were stained with the Vybrant DiI Cell-Labeling Solution (Invitrogen, Montigny-le-Bretonneux, France) for 1 h at 37 °C. Labeled cells were seeded in cell culture inserts (Corning, Boulogne Billacourt, France) with Matrigel coating (1 mg/mL, BD), at 5 × 10^4^ cell /well in RMI 1640 with FBS (1%), in the absence or presence of PROKRs antagonists (1 µM). The chemoattractant (25% serum) is added to the basal chamber. Cell invasion was stopped after 24 h at 37 °C, 5% CO_2_. The chambers were then removed and fixed with paraformaldehyde and preserved in Fluorsave (Millipore, Molsheim France). The membranes of the chambers were excised and placed on to glass slides. The cells that invaded the chamber were visualized under the microscope and counted. Images were taken using Zeiss AxioVision microscope, processed using AxioVision SE64 Rel. 4.9.1 Data are expressed as percentage of invading cells compared to the untreated control condition.

### 2.10. Statistical Analysis

All statistical analyses were performed using SigmaStat (Jandel Scientific Software, SanRafael, CA, USA). Data were analyzed using one-way ANOVA. All data were verified for normality and equal variance. When normality failed, a nonparametric test followed by Dunn’s was used. (SigmaPlot and SigmaStat, Jandel Scientific Software). All data are expressed as means ± SEM (*p* < 0.001, 0.01, and 0.05).

## 3. Results

### 3.1. Effects of Combined and Independent Treatments by PROKR1 (PC7) and PROKR2 (PKRA) Antagonists on the Litter Size, Feto-Placental Weights and Placental Efficiency

To determine the effect of PC7 and PKRA on gestational outcome, we compared their effects on the litter size, placental and fetal weights and placental efficiency in all gravid mice. Treatment with PC7 or PKRA did not affect the litter size when used independently (Figure 2A), however, we observed a trend to a decrease in the litter size when antagonists were combined (Appendix A). Treatment with PC7 increased both placental and fetal weights by 17% and 14%, respectively and PKRA treatment increased placental weight by 12% and did not affect fetal weight (Figure 2B,C). Whereas, combined treatments decreased both placental and fetal weights (Appendix A). Independent treatments did not affect placental efficiency (Figure 2D); however, this parameter was significantly decreased when the antagonists were combined (Appendix A). Because the combined treatment of both antagonists decreased placental and fetal weights and placental efficiency and caused a trend to a decrease in the litter size, we further focused the study on the analyses of the independent effects of PC7 and PKRA, as they did not cause any harmful effects on the gestational outcomes. Their independent use rather increased placental and fetal weights.

### 3.2. Effects of PC7 and PKRA Antagonists on Placental Growth and Vascularization 

The increase in the placental weight of the PC7 and PKRA treated mice strongly suggested an increase in the branching and vascularization of the placenta and in the proliferation of trophoblast cells. To verify this hypothesis, we first compared the levels of expression of the mRNA of *Gcm1*, a protein involved in placental branching during early gestation; and mRNA of *Cd31*, a marker of endothelial cells. Figure 3A shows a significant increase in the expression of *Gcm1* mRNA in PC7 treated mice and Figure 3B shows an increase in *Cd31* mRNA expression in the PKRA treated mice. We then compared the levels of the PCNA protein expression, a marker of cell proliferation. Figure 3C reports a representative Western blot that compares PCNA levels in CTL, PC7 and PKRA placentas. A significant increase in cell proliferation was observed both in PC7 and PKRA placentas, Figure 3D. Analysis of the effect of the antagonists on the placental vascularization was further examined using an in situ marker of endothelial cells, the podocalyxin, Figure 3E. There was an increase in the vascularization of the labyrinth in PC7 and PKRA placentas. This increase was validated at the protein expression of CD31 using the Western blot analysis, Figure 3F,G. A significant increase in CD31 protein was observed following both antagonists treatment. These results strongly suggest that an independent antagonization of PROKR1 and PROKR2 lead to increased placental proliferation and vascularization that may explain its increased weight. Analyses of placentas collected from mice treated with both PC7 and PKRA did not show any significant differences in the levels of the mRNA of *Gcm1* and *Cd31*, Appendix A, however significant decreases were observed in the protein levels of CD31 and PCNA (Appendix A).

### 3.3. Effects of PC7 and PKRA Antagonists on Placental Structure

To better characterize the effect of PC7 and PKRA on the placental morphology and structure, we compared the decidua, the junctional zone and the labyrinth sizes in CTL, PC7 and PKRA placentas. Figure 4A shows that only placentas from PC7 treated dams exhibited a significant decrease in the size of the decidual compartment. However, there was no effect of PC7 and PKRA treatments on the size of the labyrinth and the junctional zones. Analyses of placentas collected from mice treated with the combination of both PC7 and PKRA did not show any significant difference in placental structures when compared to the control group (Appendix A).

### 3.4. Effects of PC7 and PKRA Antagonists on Trophoblast Invasion

Because trophoblast invasion leads to the establishment of the materno-fetal circulation allowing the oxygenation of the placenta, we determined the effects of PC7 and PKRA treatments on the interstitial trophoblast invasion that is mediated by glycogen cells in the rodents. Figure 4B depicts microphotographs of the representative placentas stained for glycogen cells. There were significant increases in the number of invasive trophoblast cells within the decidual tissues collected from PC7 and PKRA treated dams, Figure 4C. This was concomitant with a significant decrease in the expression of the marker of tissue hypoxia, the CA9 protein; following PKRA treatment, Figure 4D,E. These results suggest that the independent treatments reversed the previously reported effects of EG-VEGF on the control of premature of trophoblast invasion [8,29,31] and that this effects was more pronounced upon PROKR2 antagonization. Analyses of placentas collected from mice treated with both PC7 and PKRA also showed an increase in the number of invasive cells within the decidual tissues (Appendix A). This was concomitant with a trend to a decrease in the expression of CA9 protein, Appendix A.

### 3.5. Effects of PC7 and PKRA Antagonists on Key Trophoblast Developmental Genes

To determine whether the treatments with the antagonists affected the expression of key placental genes, especially those involved in the control trophoblast differentiation and invasion, we compared the mRNA levels of *Mash2*, a gene required for the maintenance of TGC (Trophoblast giant cell) precursors; *Hand 1*, a gene that promotes TGC terminal differentiation; *Pl 1* (placental lactogen1) & 2, two TGC secreted hormones by stem cells during early and mid-gestation, respectively; and Proliferin, another TGC secreted hormone with paracrine effects on maternal endothelial cells with a significant role in vascular remodeling [46]. Figure 5 shows that neither PC7 nor PKRA affected the mRNA expression of the five genes examined. However, a trend towards a decrease in *Mash2* (panel A) and increase in *Hand1* (panel B) and *PL2* (panel E) expression were observed, suggesting an overall trend to an enhanced trophoblast differentiation. Analyses of the placentas collected from mice treated with combine PC7 and PKRA did not show any significant difference in *Hand1, Mash2* and *Pl2* mRNA expression compared to the placentas collected from the control groups (Appendix A).

### 3.6. Effects of PC7 and PKRA Antagonists on TGC Differentiation, Migration and Invasion in an In Vitro System

The in vivo analysis strongly suggested that both PC7 and PKRA treatments affected trophoblast differentiation of glycogen cells. Because in rodents, trophoblast invasion proceeds along two pathways; the interstitial trophoblast invasion that involves glycogen cells, and the endovascular trophoblast invasion that involves TGC; we further characterized the effects of the antagonists on TGC differentiation using an in vitro model system. We used the RCHO-1 cell line which was maintained in a proliferative or differentiated state [42,43]. Firstly, we examined the effects of PC7 and PKRA on the cellular morphology of RCHO-1. Figure 6A shows that the independent treatments of the RCHO-1 cells with either PC7 or PKRA enhanced their differentiation potential towards giant trophoblast cell formation. Differentiated giant trophoblast cells are depicted on the photomicrographs by the dotted lines. Combined treatment caused a slight change in the morphology of RCHO-1 cells as depicted on the photomicrograph in panel (D). To further characterize PC7 and PKRA mediated effects on the differentiation of RCHO-1 cells, we determined their effects on the mRNA expression of *Hand 1, Pl2* and *Mash2*. Figure 6B,C show that the treatment of RCHO-1 cells with PC7 or PKRA alone, or in combination did not affect the expression of *Hand 1* and *Pl2*. A trend to a decrease was observed for the *Mash2* gene with the combined treatment (Figure 6D), suggesting a potential effect of both antagonists on the differentiation of RCHO-1 cells.

Secondly, we determined the effects of independent and combined treatments on the migration and invasion of RCHO-1 cells. Figure 7A depicts photomicrographs of the migration of RCHO-1 cells at 0 (T0 h) and 12 h (T12 h) following wound formation. The cells were treated at T0 with PC7 (1 µM), PKRA (1 µM) or both. Figure 7B depicts the percentages of wound closure following 12 h of treatment. There were trends to increase in RCHO-1 migration in the presence of PC7 or PKRA. A significant decrease in RCHO-1 migration was observed when antagonists were combined compared to PC7 or PKRA alone. We then determined the effect of both antagonists on RCHO-1 cell invasion. Figure 7C, shows representative photomicrographs of RCHO-1 invasion in the presence or absence of PC7, PKRA or treatment of cells with the combination of both. As shown, there was an increase in the number of cells that invaded the matrigel following treatment with PC7. No change was observed when the cells were treated with PKRA or the combination of both antagonists compared to control cells, Figure 7D.

## 4. Discussion

Using PROKR1 and PROKR2 antagonists during early pregnancy, we confirmed the role of EG-VEGF in the control of trophoblast invasion and placental development and provide a proof of concept study for their potential use to reverse EG-VEGF-mediated adverse effects in pregnancy pathologies.

To date, the Evidence for EG-VEGF control in the key processes of placental development were mainly based on in vitro studies [4,20,29]. This work represents the first proof-of-concept study for the significant role of the prokineticin in an in vivo model and has provided evidence for the safe of use of PROKR antagonists during pregnancy. While, the combination treatment by both antagonists highlighted the importance of EG-VEGF signaling during early pregnancy; the independent treatment that employed either PC7 or PKRA provided the significant role of the two receptors, PROKR1 and PROKR2 mediated-EG-VEGF effects on gestational outcomes.

Our study also reports that the combination treatment of both antagonists led not only to a decrease in placental and fetal weight but also demonstrated a decrease in the litter size. This finding highlights the direct involvement of EG-VEGF in the growth of the placenta and in the success of the pregnancy outcome. Furthermore, the combination treatment also enhanced both trophoblast invasion and differentiation, the two key aspects that are controlled by EG-VEGF during early pregnancy to prevent premature differentiation and trophoblast invasion [4,20,29].

Our study is the first to report the significant role of EG-VEGF in a normal pregnancy using an in vivo model system. The independent use of either PC7 or PKRA alone in the in vivo model system show their therapeutic potential in increasing feto-placental weight in pregnancy pathologies associated with aberrant EG-VEGF signaling. Our study also, suggests that both receptors may directly or indirectly be involved in the EG-VEGF-mediated trophic effects of the placenta, and that when one receptor is antagonized, the other may compensate and enhance EG-VEGF mediated effects on the feto-placental growth. This assumption needs however to be verified. 

Our findings also report a decrease in both fetal and placental weights following the combination treatment with both antagonists. This is in agreement with observations, where the inhibition of prokineticin receptors was demonstrated to cause adverse effects on cell survival [47,48].

PROKR1 and PROKR2 involvement in trophoblast invasion was further substantiated by the observations reported in this study using the rodent RCHO-1 cell, as the number of invasive cells was increased upon the treatment with either PC7 or PKRA. It was also noted that PROKR2 antagonization led to a decrease in CA9 with a concomitant increase in the number of invasive cells, which suggests an EG-VEGF-mediated control of invasion is predominantly regulated by its signaling through the PROKR2 receptor [11,12,18,20,30,31]. This was further substantiated by the selective trend to a decrease in *Mash2* gene associated with a trend to an increase in *Hand1* gene, two transcription factors that are essential to maintain and promote giant cell differentiation [46].

In this study, the invasion potential of both endovascular and interstitial trophoblasts was investigated using the RCHO-1 cell in vitro and the in vivo system, respectively. In both systems PC7 and PKRA significantly enhanced trophoblast invasion. More interestingly, it was observed that the endovascular invasion was more pronounced when PROKR2 antagonist alone was used. This suggests that EG-VEGF signaling on giant cell differentiation could directly or indirectly be mediated by a PROKR2 dependent mechanism.

Altogether, these data demonstrate that PROKR1 and PROKR2 may exhibit differential roles when activated. EG-VEGF activation of PROKR1 may control placental vascularization and interstitial trophoblast invasion, and PROKR2 activation may mediate trophoblast differentiation towards giant cells and their invasion of the maternal vascular system. 

PROKR1 and PROKR2 receptors share 87% homology in their amino acid sequence, which may suppose similar activation mechanisms for the two receptors; however a difference in the final cellular response in a cell type that expresses both PROKRs, might also depend on the repertoires of G proteins present in each cell type. Importantly, we have recently demonstrated that G-protein concentrations in human trophoblast endothelial cells may influence the selectivity of coupling to PROK receptors [20]. This is line with studies that demonstrated that in living cells the expression levels of specific G-protein subunits may regulate receptor-coupling preferences [49].

For the clinical therapeutic perspectives, our findings strongly support the prospect of use of the independent antagonists rather than the combination of both PROKR1 and PROKR2 antagonists to treat pathologies that are associated with aberrant EG-VEGF expression [20,27,28,31,38,50,51,52]. In the context of pregnancy, EG-VEGF levels have been reported to be increased in numerous pregnancy pathologies, including ectopic pregnancies, preeclampsia, FGR, gestational choriocarcinoma as well as those associated with inflammation [11,12,27,28,30,31,32,53,54,55]. Thus, this proof-of concept study provides new insights into the safe use of the PROKR1 and PROKR2 receptor antagonists during early pregnancy. PROKRs antagonisation could also be considered in combination with the antagonisation of circulating EG-VEGF or VEGF. Importantly, Kaur et al. [56,57] have highlighted the importance on the use of aptamer mediated antagonism of VEGFA signalling action in angiogenesis. For example, pegaptanib is a pegylated 27 nucleotide RNA aptamer which specifically binds to the predominant 165 isoform of VEGFA, blocks its interaction with the cognate receptor, and arrests blood vessel growth [56,57].

In this study, the in vivo model system was treated until E10.5, the choice of this period of treatment was based on the observations, in which EG-VEGF levels in normal gravid mice was reported to be at maximum at 10.5 and declined thereafter [12,22]; therefore, treatment beyond E10.5 was not implemented in this study. The beneficial effect of the use of the antagonists in the pregnancy context, is further supported by the numerous studies that demonstrated their strong effects on the alleviation of pain and inflammation [58,59,60,61]. 

Although the doses and the mode of delivery of the therapeutic agents via systemic delivery/subcutaneous injections have proven tolerance and preservation of the gestation, one of the major limitations on the dosage and the mode of delivery of the antagonists agents is that they have not been delivered specifically in the placenta. Nevertheless, the doses and duration of the treatment used in this study were based on strong previous studies from other groups who used either PROKR1 or PROKR2 antagonists to treat pain and inflammatory reactions that were reported to be associated with an increase in the circulating or local levels of prokineticins (PROK1 or PROK2) and their receptors [15,16,25,28,36,37,44,45,62].

Altogether, these data provide the potential use of PROK1 and PROK2 antagonists in the treatment of pathologies associated with an aberrant EG-VEGF-signaling, including those related to pregnancy [13,34,35,36,37,63].

## 5. Conclusions

In conclusion, this study has demonstrated the key receptor(s) dependent mechanisms underlying EG-VEGF-mediated signaling during early pregnancy and provides a proof concept for the potential therapeutic use of the antagonists of its receptors to treat pregnancy pathologies in a murine model of normal pregnancy.

## Figures and Tables

**Figure 1 biomedicines-09-00309-f001:**
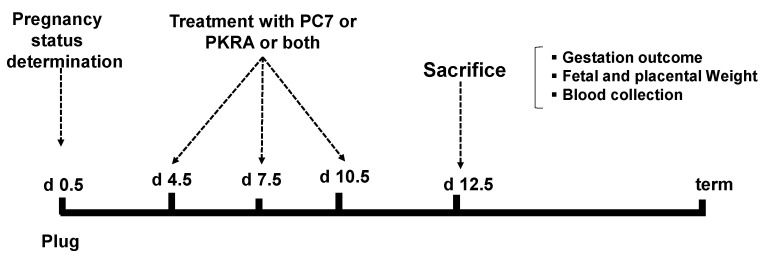
Experimental procedure. The figure illustrates the flow chart of the experimental procedure performed at different time-points during the study. The gravid mice were randomly assigned to be injected with either vehicle, PROKR1 antagonist (PC7) or PROKR2 antagonist (PKRA), or both (PC7+PKRA). The treatment with antagonists started on day 4.5 of gestation and were repeated every 3 days, at 7.5 dpc; 10.5 dpc. Mice were sacrificed at 12.5 dpc.

**Figure 2 biomedicines-09-00309-f002:**
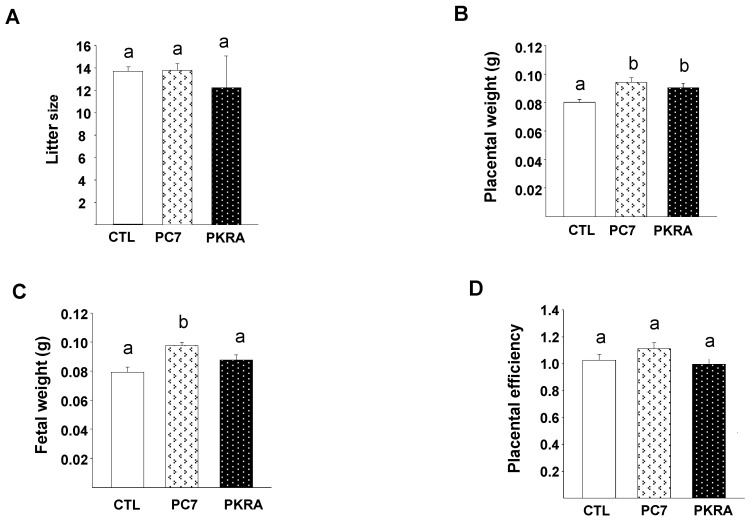
Effects of prokineticin antagonists’ PC7 or PKRA on gestation outcomes. (**A**) depicts a graph that compares the litter size of gravid mice treated or not by PC7 or PKR-A. (**B**,**C**) report graphs that compare placenta and fetal weights of pups born from mother treated or not by PC7 or PKRA, respectively. (**D**) reports graph that compare placental efficiency of mice treated or not by PC7 or PKRA. Data are presented as mean ± SEM. *p* < 0.05. Bars with different letters are significantly different from each other.

**Figure 3 biomedicines-09-00309-f003:**
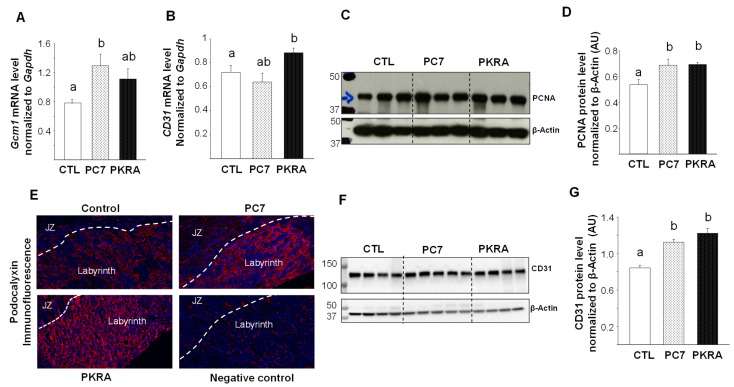
Effects of prokineticin antagonists’ on placental growth and vascularization. (**A**,**B**) depict comparisons of *Gcm1* and *Cd31* mRNA levels in placentas of CTL; PC7 and PKRA, respectively. *Gapdh* was used to standardize for mRNA expression *p* < 0.05. (**C**,**D**) shows Western blot analysis and quantification of CD31 protein levels in the placenta collected from CTL, PC7 and PKRA treated mice. Standardization of immunoreactivity was performed using antibodies against β-actin. Data are represented as mean ± SEM. *p* < 0.05. (**E**) shows representative immunofluorescence photomicrographs of placental sections stained for Podocalyxin (red staining) merged with Hoechst (Blue). Placentas were collected from CTL, PC7 and PKRA treated mice at 12.5 dpc. JZ: Junctional Zone. (**F**,**G**) show western blot analysis and quantification of CD31 protein levels in the placenta collected from CTL, PC7 and PKRA treated mice. Standardization of immunoreactivity was performed using antibodies against β-actin. *p* < 0.05. Bars with different letters are significantly different from each other.

**Figure 4 biomedicines-09-00309-f004:**
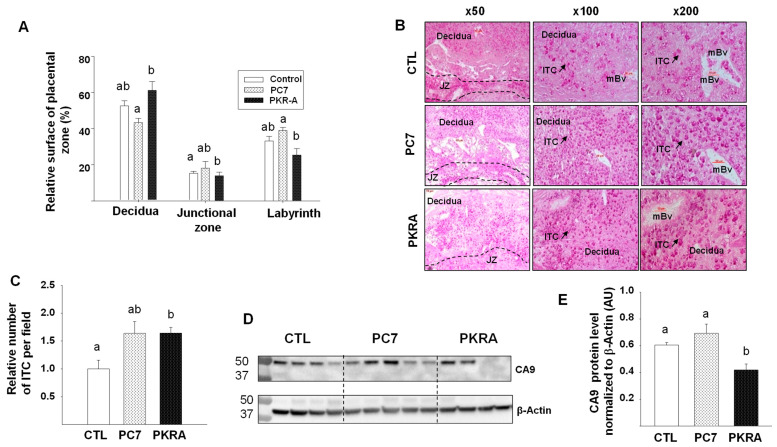
Effects of prokineticin antagonists’ on placental structure and trophoblasts invasion. (**A**) depicts analysis of the placental zones of CTL, PC7 and PKRA placentas. For each group, at least three placental sections/animal were analyzed. The graph shows proportions of the surface layer of 3 zones of the placenta (labyrinth, junctional zone, and decidua). Surfaces of the 3 layers were measured on parasagittal sections for each placenta. Mean values were used to calculate the mean (SEM) surface proportion of the layers. *p* < 0.05. NS, not significant. (**B**) shows representative photomicrographs of placental sections stained with Peroxide Acid Shiff at different magnifications. Placentas were collected from CTL, PC7 and PKRA treated mice at 12.5 dpc. JZ: Junctional Zone; ITC: Invasive Trophoblast Cell; mBv: Maternal Blood Vessel. (**C**) depicts a graph that compares the number glycogenic cells in the maternal decidua. (**D**,**E**) shows western blot analysis and quantification of CA9 protein levels in the placenta collected from CTL, PC7 and PKRA treated mice. Standardization of immunoreactivity was performed using antibodies against β-actin. Data are represented as mean ± SEM. *p* < 0.05. Bars with different letters are significantly different from each other.

**Figure 5 biomedicines-09-00309-f005:**
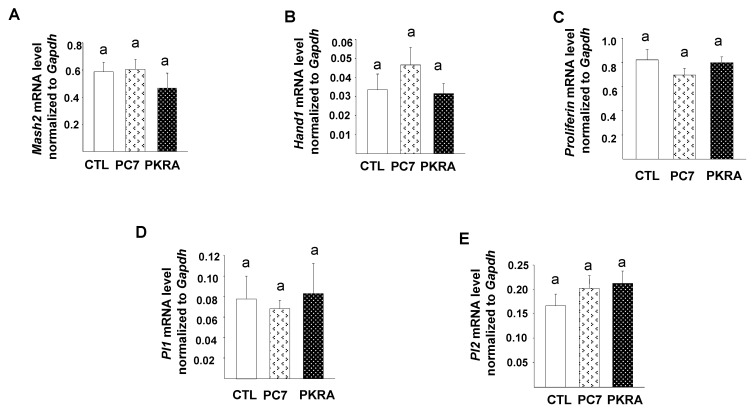
Effects of prokineticin antagonists’ treatment on key placental development genes expression. (**A**–**E**) depict comparisons of *Mash2*, *Hand1*, *Proliferin* and *Placental lactogen1* and *2* mRNA levels, respectively in placentas of collected from CTL; PC7 and PKRA treated mice. *Gapdh* was used to standardize for mRNA expression. Data are represented as mean ± SEM. *p* < 0.05. Bars with different letters are significantly different from each other.

**Figure 6 biomedicines-09-00309-f006:**
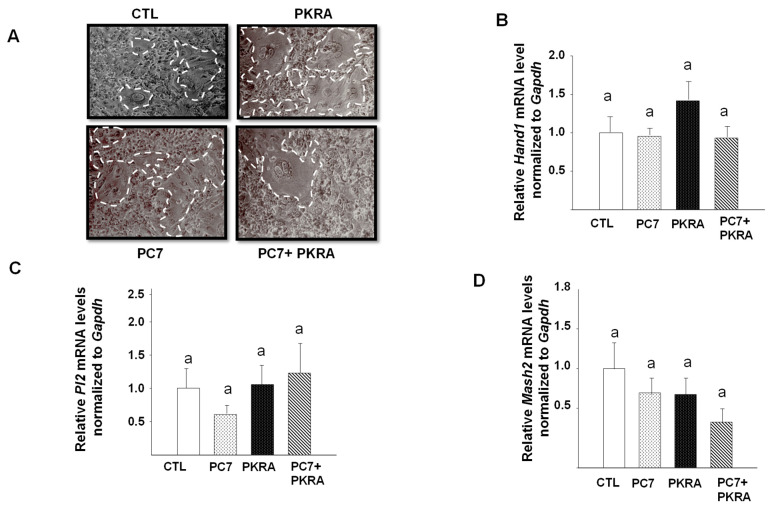
Effects of prokineticin antagonists’ on RCHO-1 differentiation. (**A**) reports representative microphotographs of RCHO-1 cells in the absence or presence of PC7 (1 µM), PKRA (1 µM), or both. (**B**–**D**) depict comparisons of, *Hand1*, *Placental lactogen2* and *Mash2* mRNA levels, respectively on total extracts fromRCHO-1 treated or not with PC7 or PKRA or both. *Gapdh* was used to standardize for mRNA expression. Data are represented as mean ± SEM of three independent experiments. *p* < 0.05. Bars with different letters are significantly different from each other.

**Figure 7 biomedicines-09-00309-f007:**
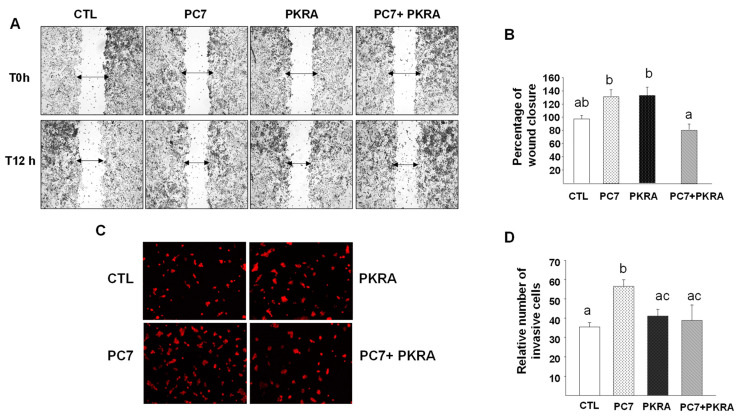
Effect of prokineticin antagonists’ on RCHO-1 migration and invasion. (**A**) depicts photographs of wounded RCHO-1 cells at 0 (T0) and 12 h (T12 h) post-wounding. The cells were treated at T0 with PC7 (1 µM) or PKRA (1 µM) or both. (**B**) shows a graph that reports the percentages of wound closure after 12 h of treatment. (**C**) reports microphotographs of RCHO-1 cells prelabeled with acetylated low-density lipoprotein (Dill) cultured on Matrigel-coated HTS fluroBlok transwell inserts and treated or not with antagonists. (**D**) depicts the % of invading cells upon treatment with PC7 (1 µM) or PKRA (1 µM) or both. Data are presented as means ± SEM. *p* < 0.05. Bars with different letters are significantly different from each other.

**Table 1 biomedicines-09-00309-t001:** Shows the list of the primers used to perform q-PCR in the study.

Genes	Forward	Reverse	Amplicon Size	Tm (°C)
**m*Gapdh***	AACGACCCCTTCATTGAC	TCCACGACATACTCAGCA	191	57
**m*Mash 2***	GGTGACTCCTGGTGGACCTA	TCCGGAAGATGGAAGATGTC	151	56
**m*Pl1***	TGTCATACTGCTTCCATCCATAC	CCAGGTGTTTCAGAGGTTCTT	125	60
**m*Pl2***	ACGCCCATGATCTTGCTTCA	TGGCAGGGGCTTAACATCAG	114	60
**m*Proliferin***	TGTGTGCAATGAGGAATGGT	TAGTGTGTGAGCCTGGCTTG	223	58
**m*Gcm1***	TTTTTCCAGTCCAAAGGCGAG	TGACTCGGGATTTCAGCAGG	164	60
**m*Cd31***	GCATCGGCAAAGTGGTCAA	GTTCCATTTTCGGACTGG C	145	60
**m*Hand1***	AAGCAAGCGGAAAAGGAGT	GGCCTGGTCTCACTGGTTTA	159	60

## Data Availability

Not applicable.

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
