# Peer review of "Evidence-Based View of Safety and Effectiveness of Prokineticin Receptors Antagonists during Pregnancy"

_biomedicines, 2021, doi:10.3390/biomedicines9030309_

Round 1

Reviewer 1 Report

The study is well thought-out and scientifically sound. I would like authors to add bit of information related to VEGF along with references.

Line 44: The author must add 2-3 sentences about VEGF, its function and its antagonist. Also, the author must add the following references.

  • Kaur H, Li JJ, Bay BH, Yung LY. Investigating the antiproliferative activity of high affinity DNA aptamer on cancer cells. PLoS One. 2013;8(1):e50964.
  • Ruckman J, Green LS, Beeson J, Waugh S, Gillette WL, et al. (1998) 2’-fluoropyrimidine RNA-based aptamers to the 165-amino acid form of vascular endothelial growth factor (VEGF165): Inhibition of receptor binding and VEGF-induced vascular permeability through interactions requiring the exon 7-encoded domain. J Biol Chem 273: 20556–20567.
  • Kaur H, Yung LY. Probing high affinity sequences of DNA aptamer against VEGF165. PLoS One. 2012;7(2):e31196.

Author Response

Reviewer 1

Question 1

The study is well thought-out and scientifically sound. I would like authors to add bit of information related to VEGF along with references.

Line 44: The author must add 2-3 sentences about VEGF, its function and its antagonist. Also, the author must add the following references.

  • Kaur H, Li JJ, Bay BH, Yung LY. Investigating the antiproliferative activity of high affinity DNA aptamer on cancer cells. PLoS One. 2013;8(1):e50964.
  • Ruckman J, Green LS, Beeson J, Waugh S, Gillette WL, et al. (1998) 2’-fluoropyrimidine RNA-based aptamers to the 165-amino acid form of vascular endothelial growth factor (VEGF165): Inhibition of receptor binding and VEGF-induced vascular permeability through interactions requiring the exon 7-encoded domain. J Biol Chem 273: 20556–20567.
  • Kaur H, Yung LY. Probing high affinity sequences of DNA aptamer against VEGF165. PLoS One. 2012;7(2):e31196.

Author’s response to reviewer 1

We acknowledge this reviewer's comment and thank him (her) for his (her) insightful comments. We have now added the following paragraphs and suggested references where appropriate.

The following sentence has been added line 47-49 in the Introduction section

While VEGF has mainly been reported to play a crucial role in promoting angiogenesis with specific and strong mitogenic and chemotactic actions on endothelial cells and to induce microvascular permeability (Robinson CJ and Stringer SE, 2001), prokineticins act as both pro-angiogenic and pro-inflammatory factors.

The following sentence has been added line 445-450 in the Discussion section

PROKRs antagonisation could also be considered in combination with the antagonisation of circulating EG-VEGF or VEGF. Importantly, Kaur et al [56,57] have highlighted the importance on the use of aptamer mediated antagonism of VEGFA signaling action in angiogenesis. For example, pegaptanib is a pegylated 27 nucleotides RNA aptamer which specifically binds to the predominant 165 isoform of VEGFA, blocks its interaction with the cognate receptor, and arrests blood vessel growth [56,57].

Reviewer 2 Report

In this work, the authors claimed that they presented the first proof of concept for the important role of PROKR in the early stages of pregnancy and evidence to safely use of PROKR antagonists during pregnancy. However, despite the various meaningful findings, I am deeply concerned about significant defects in the design and conduct of this study in the aspect of developmental and reproductive toxicology.

The authors subcutaneously administered PC7 and PKRA as PROKR antagonists at 0.5 and 75 mpk, respectively, on 4.5, 7.5 and 12.5 dpc, and examined the gestational outcomes, placental growth, and other parameters. However, neither PC7 nor PKRA's references cited by the authors provide any basis for in vivo exposure by subcutaneous injection. Furthermore, it has not been sufficiently examined whether the dose and the dosage of the PROKR antagonists are reasonable as a toxicology study. These weaknesses present fundamental limitations that are hard to agree with all the findings and arguments presented in this study.

Despite the interesting results of the therapeutic potential of gestational defects through PROKR antagonism, it is difficult to publish the current state of the work unless these limitations are properly addressed.

Author Response

Reviewer 2

Question 1

In this work, the authors claimed that they presented the first proof of concept for the important role of PROKR in the early stages of pregnancy and evidence to safely use of PROKR antagonists during pregnancy. However, despite the various meaningful findings, I am deeply concerned about significant defects in the design and conduct of this study in the aspect of developmental and reproductive toxicology.

The authors subcutaneously administered PC7 and PKRA as PROKR antagonists at 0.5 and 75 mpk, respectively, on 4.5, 7.5 and 12.5 dpc, and examined the gestational outcomes, placental growth, and other parameters. However, neither PC7 nor PKRA's references cited by the authors provide any basis for in vivo exposure by subcutaneous injection. Furthermore, it has not been sufficiently examined whether the dose and the dosage of the PROKR antagonists are reasonable as a toxicology study. These weaknesses present fundamental limitations that are hard to agree with all the findings and arguments presented in this study.

Despite the interesting results of the therapeutic potential of gestational defects through PROKR antagonism, it is difficult to publish the current state of the work unless these limitations are properly addressed.

Author’s response to reviewer 2

We acknowledge this reviewer's comment and thank him (her) for the insightful comments to substantially improve the revised manuscript.

Although the doses and the mode of delivery of the therapeutic agents via systemic delivery/subcutaneous injections have proven tolerance and preservation of the gestation (estimated by the absence of changes in the litter size when one or the other antagonist has been used), one of the major limitations on the dosage and the mode of delivery of the antagonists agents is that they have not been delivered specifically to the placenta. The doses and duration of the treatment used in this study were based on strong previous studies from other groups who used either PROKR1 or PROKR2 antagonists to treat pain and inflammatory reactions that were reported to be associated with an increase in the circulating or local levels of prokineticins (PROK1 or PROK2) and their receptors.

The following paragraphs report a non-exhaustive summary of references from previous studies that used PROKR1 or PROKR2 antagonists. Please note that our study is the first to test these molecules in pregnancy setting. We believe that this will be a starting point for future studies that will investigate the local delivery of these molecules to the placenta.

PROKR1 antagonist, PC1 or PC7:

In vitro studies:

  • Goryszewska E et al, Biology of reproduction, 2021
  • Moschetti G et al, Front Immunol, 2020;
  • Caioli S et al, Neuropharmacology, 2017
  • Traboulsi W et al, Clin Cancer Res, 2017
  • Garnier V et al, Am J Physiol Endocrinol Metb, 2015
  • Congiu C et al, Eur J Med Chem, 2014

In vivo studies:

  • Moschetti et al Brain Behav Immun, 2019.
  • Maftei et al, Neuropharmacology, 2019
  • Moschetti G et al, J Neuroinlammation, 2019
  • Castelli M, Plos One, 2016
  • Lattanzi R et al, Biomed Res Int, 2015
  • Guida F et al, Pharmacol Res, 2015
  • Maftei et al BR J Pharmacol, 2014
  • Traboulsi W et al, Clin Cancer Res, 2017

PROKR2 antagonist, PKRA:

In vitro studies:

  • Ren C et al, Eur J Pharmacol, 2015
  • Garnier V et al, Am J Physiol Endocrinol Metb, 2015
  • Traboulsi W et al, Clin Cancer Res, 2017

In vivo studies:

  • Chen B et al, Mediators Inflamm, 2015
  • Zhou, qy et al, Mol Brain, 2016
  • Xia L et al, Plos one, 2014
  • Cheng MY et al , Proc Natl Acd Sci, 2012
  • Traboulsi W et al, Clin Cancer Res, 2017

All the studies that used the antagonists developed by Dr Balboni (PROKR1 anatgonist) or by Dr Zhou (PROKR2 antagonist) have largely used these antagonists in in vivo studies (see above references) and at similar doses or even higher, when considering the frequency of injections. The duration used by all these groups has been adapted to the considered pathology. In our case, the doses and duration have been used within the first period of gestation, during this period EG-VEGF exhibits the highest levels of expression and secretion, in mice and in women. One of the main objectives of this first study was to demonstrate the safety-based utilization of these molecules during normal gestation in order to consider their use in non-tumoral pregnancies, such as preeclampsia and fetal growth restriction. Please note that the same doses have been used in our publication in 2017 that was mainly focused on the pathology of gestational choriocarcinoma (a rare cancer of pregnancy) (Traboulsi et al 2017). In this study, we were pleased to observe that these antagonists have reduced tumor growth and preserved gestation.

As reported in the objectives of this study, these molecules have first been used to demonstrate the role of EG-VEGF in vivo and, if safe to be considered in pregnancy associated pathologies. It is to be noted that this study allowed abandoning the possibility of using both antagonists at the same time as this may affect the pregnancy outcome.

We have now added a paragraph in the discussion section line 457-464 that reports the limitations of our study and the future directions in regards to the use of prokineticin antagonists to treat prokineticin-related pathologies, especially those developed during pregnancy. New references in support of the safety of PROKR1 and PROKR2 antagonist use in vivo have now been added.

Reviewer 3 Report

EG-VEGF acting via the two G-protein coupled receptors, PROKR1 and PROKR2 contributes to placental vascularization and growth. EG-VEGF is highly expressed in the human placenta and aberrant expression is associated with pregnancy pathologies. In this study authors analyze in vivo the distinct role of prokineticin receptors using specific antagonists PC7 (PROKR1) and PKRA (PROKR2), reported to reverse PROKs adverse effects in other system The data demonstrate that PROKR1 and PROKR2 may exhibit differential roles: EG-VEGF activation of PROKR1 may control placental vascularization and interstitial trophoblast invasion, and PROKR2 activation may mediate trophoblast differentiation towards giant cells and their invasion of the maternal vascular  system. The study seems particularly interesting suggesting the possibility of modulating the expression of placental key transcription factors by blocking the signal of prokineticins.

The paper is well written, the experiments are convincing and well designed. I suggest that it would be worthwhile also to examine  the cascades that determine a different effect following the activation of the two receptors. However this elegant study suggests the possibility of selectivity antagonization of prokineticin receptors may constitute a potential therapy for the pregnancy pathologies.

Author Response

Reviewer 3

Comments and Suggestions for Authors

EG-VEGF acting via the two G-protein coupled receptors, PROKR1 and PROKR2 contributes to placental vascularization and growth. EG-VEGF is highly expressed in the human placenta and aberrant expression is associated with pregnancy pathologies. In this study authors analyze in vivo the distinct role of prokineticin receptors using specific antagonists PC7 (PROKR1) and PKRA (PROKR2), reported to reverse PROKs adverse effects in other system The data demonstrate that PROKR1 and PROKR2 may exhibit differential roles: EG-VEGF activation of PROKR1 may control placental vascularization and interstitial trophoblast invasion, and PROKR2 activation may mediate trophoblast differentiation towards giant cells and their invasion of the maternal vascular  system. The study seems particularly interesting suggesting the possibility of modulating the expression of placental key transcription factors by blocking the signal of prokineticins.

The paper is well written, the experiments are convincing and well designed. I suggest that it would be worthwhile also to examine  the cascades that determine a different effect following the activation of the two receptors. However this elegant study suggests However this elegant study suggests the possibility of selectivity antagonization of prokineticin receptors may constitute a potential therapy for the pregnancy pathologies.

Author’s response to reviewer 3

We thank the reviewer for his (her) insightful and constructive comments.

As emphasized by the reviewer the present study supports the idea that selective, rather than simultaneous antagonization of prokineticin receptors may constitute a potential therapy for the pregnancy pathologies. Nevertheless, the present in vivo study could not allow differentiating the specific mediated PROKR1 and PROKR2 effects, as these receptors are expressed by different cells types in the placentas, including trophoblast, endothelial and hoffbauer cells.

In previous studies from our group using either endothelial cells isolated from the human placenta (Brouillet et al) or trophoblast cells (primary or explants cultures), we could differentiate major mediated effects upon the activation of PROKR1 or PROKR2 receptors. In those studies, we used siRNA strategy; blocking antibodies or PC7/ PRKRA antagonists and demonstrated that in vitro, PROKR1 activation mediated more the cell’s proliferation and that PROKR2 was more associated with cell’s permeability and differentiation. Importantly in the study that used endothelial cells (Macrovascular (HUVEC) (human umbilical vascular endothelial cells) or microvascular placental cells (Microvascular (HPEC) human placental endothelial cells), we demonstrated that PROK-receptors signaling exhibited differential response to EG-VEGF.

The differential response of HUVEC and HPEC to EG-VEGF might be due to the repertoire of G protein expressed in each of the cell types. In fact, HPEC cells express three times more Gαi2 and three times less Gαi1 compared with HUVECs. The higher levels of Gαi1 over Gαi2 has been previously reported at the protein levels in HUVEC cells (Masri et al., 2006). Those results were in line with studies that showed that the degree of inhibition of adenylyl cyclase was higher in cells expressing Gαi2 than in cells expressing Gαi1 (Massotte et al., 2002; Masri et al., 2006).

PROKR1 and PROKR2 receptors share 87% homology in their amino acid sequence, which may potentiate similar activation mechanisms for the two receptors; however, a difference in the final cellular response in a cell type that expresses both PROKRs, might also depend on the repertoires of G proteins present in each of the cell types. It is now well documented that the selectivity of coupling depends on the G-protein concentration in a given cell (slessareva et al 2003). This suggests that in living cells the expression levels of specific G-protein subunits may regulate receptor-coupling preferences. In addition, we observed that HPEC and HUVEC cells express different Gαi proteins. This may allow these cells to perform different physiological functions in response to stimulation by the same ligand.

These precisions are now added in the discussion section, line 429-436.

Reviewer 4 Report

The subject of the manuscript is very interesting and proper for the journal. One of main advantage of the study  is using in vivo model. The data provide rational of potential use of the PROK1 antagonists to treat pregnancy pathologies.  

The following major and minor issues require the Authors’ attention:

  • The statistical approach seems not proper. There is inconsistency in the presenting results.

The all groups: Control, PKRA, PC7, PC7+PKRA should always compared in the same way by on-way ANOVA (as it is on some graphs). On some graphs however there are separated two sets: control, PC& and PKRA (one-way ANOVA) and second set: CTR and PC7+PKRA (t-test) (as the other graphs show). This is inconsistency.

  • The newest papers concerning role of PROK1 and its receptor in pregnancy are not disused (e.g. Biol Reprod. 2021 Jan 4;104(1):181-196; Biol Reprod. 2020 Aug 21;103(3):654-668).
  • How doses of used receptor antagonists were chosen? It should be explained.
  • Did the authors determine if in their experiments GAPDH is stable gene?
  • There are some mistakes in Fig 3 legend. Description of CD and EF do not match with graphs.
  • What were the negative controls for immunofluorescence analyses and other microscopic analyses? They should be described.
  • Abbreviations should be explained in figure legends.
  • In paragraph 3.6 there are letters (b) and (c). It is not clear to what these letters refer.

Author Response

Reviewer 4

Comments and Suggestions for Authors

The subject of the manuscript is very interesting and proper for the journal. One of main advantage of the study  is using in vivo model. The data provide rational of potential use of the PROK1 antagonists to treat pregnancy pathologies.  

The following major and minor issues require the Authors’ attention:

Author’s response to reviewer 4

Question 1

  • The statistical approach seems not proper. There is inconsistency in the presenting results.

The all groups: Control, PKRA, PC7, PC7+PKRA should always compared in the same way by on-way ANOVA (as it is on some graphs). On some graphs however there are separated two sets: control, PC& and PKRA (one-way ANOVA) and second set: CTR and PC7+PKRA (t-test) (as the other graphs show). This is inconsistency.

Response 1

We apologize for any inconvenience regarding the presentation of the data and for the statistical analyses. All data have been re-analyzed using one way Anova to consider the comparisons between all groups. All figures have been reformatted in consequence. Please note that bar graphs that have different letters are different from each other.

We apologize for not providing enough clarity for the separation between the graphs that report independent treatments groups and those with the combination treatments. Because combined PC7 and PKRA were injected separately, the control mice for the combined treatments also received two injections. Hence, the controls for the independent or combined treatments were considered as different. Hence, we analyzed the mice treated with combined molecules to their proper controls. For more clarity of the data, all in vivo data with the combined molecules are reported in the supplementary material. For the independent treatments, we have grouped all control used for PC7 or PKRA in one single group of controls since there was no difference in the pregnancy outcomes was observed (see attached figure for reviewed appreciation).

Please note that for the in vitro study the same control served for all conditions, as the final working solutions, including the control, were prepared using PBS.

All these details are now added in the methods section, line 107-115 and line 186-190

Question 2

  • The newest papers concerning role of PROK1 and its receptor in pregnancy are not disused (e.g. Biol Reprod. 2021 Jan 4;104(1):181-196; Biol Reprod. 2020 Aug 21;103(3):654-668).

Response 2

We agree with the reviewer that these papers are interesting as they substantiate a significant role of EG-VEGF/PROK1 in pregnancy and also used PC1, the reference antagonist for PROKR1 receptor. These papers are discussed and cited where appropriate within the revised version the manuscript.

How doses of used receptor antagonists were chosen? It should be explained.

All the studies that used the antagonists developed by Dr Balboni (PROKR1 antagonist) or by Dr Zhou (PROKR2 antagonist) have largely used these antagonists in in vivo studies and at similar doses or even higher, when considering the frequency of injections (please see references cited to respond to reviewer’s 2 question). The duration used by all these groups has been adapted to the considered pathology. In our case, the doses and duration have been used within the first period of gestation, during which EG-VEGF exhibits the highest levels of expression and secretion, in mice and in women. One of the main objectives of this first study was to demonstrate the safety-based utilization of these molecules during normal gestation in order to consider their use in non-tumoral pregnancies, such as preeclampsia and fetal growth restriction. Please note that the same doses have been used in our publication in 2017 that was mainly focused on the pathology of gestational choriocarcinoma (a rare cancer of pregnancy) (Traboulsi et al 2017). In this study, we were pleased to observe that these antagonists have reduced tumor growth and preserved gestation.

As reported in the objectives of this study, these molecules have first been used to demonstrate the role of EG-VEGF in vivo and, if safe to be considered in pregnancy associated pathologies. It is to be noted that this study allowed abandoning the possibility of using both antagonists at the same time as this may affect the pregnancy outcome.

  • Did the authors determine if in their experiments GAPDH is stable gene?

Before performing the final PCR analyses, we performed preliminary studies using Cytokeratin7, Rpl13a, 18s rRNA and Gapdh genes as internal controls. We found that Gapdh was the most reliable and produced reproducible data. Also, recent recommendation for internal reference to be used in RT-qPCR recommends the use of gene with degree of amplification that is not two far from the candidate gene of interest and we observed that Gapdh fulfilled this criteria. Also, it was clearly reported in the literature that when considering placenta the Gapdh gene is highly recommended due to its abundance in the expression levels in a given gestation, although its expression level is shown to be varying with advanced gestation in the studies reporting on “ontogeny”. In this study all mice were compared at day 12.5 of gestation, therefore we believe the use of Gapdh as an endogenous control in our study is appropriate.

There are some mistakes in Fig 3 legend. Description of CD and EF do not match with graphs.

We thank the reviewer for this observation, we have corrected the error

  • What were the negative controls for immunofluorescence analyses and other microscopic analyses? They should be described.

Comprehensive information on the use of negative controls has now been added where appropriate

  • Abbreviations should be explained in figure legends.

All abbreviations are now clearly reported in the figure legends

  • In paragraph, 3.6 there are letters (b) and (c). It is not clear to what these letters refer.

We thank the reviewer for this observation; we have corrected the error in the revised manuscript text.

Round 2

Reviewer 4 Report

Tha authors fully adressed the reviewer's comments.